# Pigments and Techniques of Hellenistic Apulian Tomb Painting

**DOI:** 10.3390/molecules28031055

**Published:** 2023-01-20

**Authors:** Annarosa Mangone, Camilla Colombi, Giacomo Eramo, Italo Maria Muntoni, Tiziana Forleo, Lorena Carla Giannossa

**Affiliations:** 1Dipartimento di Chimica, Università degli Studi di Bari “Aldo Moro”, Via Orabona, 4, 70126 Bari, Italy; 2Centro Interdipartimentale Laboratorio di Ricerca per la Diagnostica dei Beni Culturali, Via Orabona 4, 70126 Bari, Italy; 3German Archaeological Institute, Via Sicilia 136, 00187 Roma, Italy; 4Dipartimento di Scienze della Terra e Geoambientali, Università degli Studi di Bari “Aldo Moro”, Via Orabona 4, 70126 Bari, Italy; 5Soprintendenza Archeologia, Belle Arti e Paesaggio per le Province di Barletta-Andria-Trani e Foggia, Via Alberto Valentini Alvarez, 8, 71121 Foggia, Italy

**Keywords:** pigments, madder lake, Hellenistic wall painting, plasters, Apulia

## Abstract

The wall paintings of five Hellenistic tombs in Apulia were analysed using a multi-technique approach to discern the painting techniques used and contextualise them within the phenomenon of Hellenistic tomb painting in Southern Italy. In particular, the question was asked whether technical knowledge played a role in the reception of Hellenistic artistic models and whether this knowledge was present locally. Raman and IR spectroscopies were used to identify pigments, colourants, and binders; light and electron microscopy were used to determine the structural characteristics of the paint layers and recognise the manufacturing technique. Analyses identified a fresco application for the Tomba dei Cavalieri (Arpi) and a dry application for the Canosian hypogea. The palette—typical for Hellenistic tomb painting in Southern Italy, Etruria and Macedonia—was composed of lime (white), charcoal (black), hematite (red), goethite (yellow), and Egyptian blue (blue). In the Tomba della Nike (Arpi), meanwhile, two particularly refined preparatory layers were observed. The palette was enriched with precious cinnabar and madder lake. The colouring components of the root were mixed with clay and K-alum applied on an additional layer of lime. The use of madder lake and a pink background link the painting to the polychrome Daunian pottery, and the contribution of a local workshop to the decoration of this tomb thus seems plausible.

## 1. Introduction

Investigating the manufacturing characteristics of Hellenistic items from Apulia is an extensive research project that we have been carrying out for years [1,2,3,4,5,6,7]. This paper focuses on the identification of raw materials (pigments/colorants/binders) and the verification of the painting techniques of wall painting from Hellenistic tombs in the Daunian area (northern Apulia), paying close attention to how different classes of finds relate to one another from a technological–productive perspective [8].

The phenomenon of tomb painting in ancient Apulia can be traced back to the end of the 6th and beginning of the 5th century BCE [9]. This earliest phase includes some tombs found in Peucezia and Messapia, notably a tomb with painted depictions of animals and objects from Ugento and two sarcophagus tombs found in Taranto painted with vegetal decorative elements and coloured bands. Numerous tombs with monochrome or coloured banded decoration are attested especially since the mid-5th century in Peucezia, exceptional case being the figurative decorations found in the semi-chamber graves Botromagno 1/1974 in Gravina di Puglia and in the Tomba delle Danzatrici in Ruvo [10,11]. In the 4th century BCE, and especially in the second half of the century, the phenomenon is widespread, and the walls of tombs of different types are plastered and painted. Numerous examples are known in Peucezia, where the most widespread decoration consists of coloured horizontal bands, while only occasionally themes with objects, vegetal elements or imitations of textiles and marbles are seen. Furthermore, several painted graves are attested in Messapia, where zonal decoration prevails once again, in some cases accompanied by vegetal elements, garlands, bands and weapons; the figured reliefs of the Ipogeo Palmieri in Lecce stand out as exceptional [11,12]. Starting in the last thirty years of the 4th century BCE, painted tombs are again attested in Taranto, and are mainly decorated with painted bands, sometimes combined with geometric and vegetal motifs [13]. Finally, in Daunia the phenomenon of grave painting appears in the second half of the 4th century BCE; in addition to a few examples of zonal/banded decoration, figurative friezes with narrative scenes prevail (Arpi: Tomba dei Cavalieri, Tomba del Trono, Tomba della Nike, fragments from Montarozzi; Canosa di Puglia: Ipogeo Lagrasta 1 and reliefs in the Ipogeo Monterisi-Rossignoli and Ipogeo dell’Oplita; Salapia: Ipogeo Cassano; Tiati: Tomba dei Capitelli Ionici; [12,14,15,16,17]). Over the course of the 3rd century, in the indigenous centres of Apulia the phenomenon seems to gradually lose importance. Besides the painted graves set in the previous century and still in use, fewer painted monuments are known. Again, the decorations are mostly zonal and the figurative friezes are concentrated in Daunian centres and, occasionally, in Messapia (Arpi: Ipogeo della Medusa; Canosa di Puglia: Ipogeo Sant’Aloia, Ipogeo Scocchera B, Ipogeo del Cerbero; Egnazia: Tomb Weege 49; Vaste: Ipogeo delle Cariatidi with figured relief with remains of colour [11,12]). In Taranto, however, in the 3rd and 2nd centuries BCE the phenomenon gains traction again and numerous examples of tomb painting with banded decoration, painted garlands and occasionally human figures are seen (Chamber Tombs of Piazza d’Armi, Via Gorizia and Tomb 11 of Piazza d’Armi [13]).

In general, therefore, it seems possible to state that, although tomb painting is widespread throughout Apulia, figurative and narrative decorations are mostly attested in Daunia. The review of the monuments also seems to illustrate that the impetus for the use of figurative decoration did not come from Tarentine grave paintings, which are dated largely to a later period and generally have different types of decoration. The explosion in the use of grave painting from the mid-4th century onward thus raises some questions regarding the patterns and techniques used. As widely pointed out, tomb painting from Apulia is part of a larger phenomenon that sees the spread of this art form throughout the Mediterranean during the 4th century BCE. Macedonia, with which the centres of Apulia had very close relations, is recognised as the major centre of diffusion [18,19]. If for iconographic and stylistic models, especially in terms of decorative motifs, one can thus range from Macedonia to Alexandria and from Thrace to Etruria, it is legitimate to ask what technical knowledge played a role in the adoption and reception of these models and where this knowledge came from. Can it be assumed that local craftsmen specialising in painted wall decoration already existed, or should we suppose the involvement of painters from other areas, such as Northern Greece? Or, again, is it more likely that the earliest examples of figurative tomb decoration are to be connected with local workshops of potters or other artists? Considering the differences in techniques and use of painting materials between pots and mural painting, how did the technological transfer take place? Were the same techniques and materials used for the different mediums—graves, houses and pottery—or can medium-specific differences be traced?

A brief recapitulation of wall painting techniques is necessary to enable clear interpretation of the results obtained and their correlation with the existing literature. The techniques used in mural painting are generally distinguished in fresco and secco. In fresco painting, the artist mixes water and pigment to create a paint mixture, which is then applied to wet lime plaster. The plaster absorbs the water from the paint mixture, causing the pigment to become chemically bound to the plaster through carbonation. Instead of being mixed with pure water (pure fresco), the pigment can be mixed with limewater (lime fresco), as described by Cennino Cennini [20]. The addition of a diluting binder, such as casein, as an additive will not prevent a painting from being classified as fresco, since the paint is always applied to wet plaster and the lime carbonation fixes the pigments [21].

Secco, on the other hand, is a technique in which the artist applies pigment to dry plaster or limewash. The typical binder for secco is lime milk or water (lime paint). In addition to lime paint, we find two main types of secco techniques for mural painting: tempera and oil. Tempera is a type of paint that is made by mixing the pigment with a water-soluble binder, such as egg, casein, animal glue or certain vegetable gums [22]. With respect to oil, linseed oil and poppy oil have been used for mural painting.

For both tempera and oil, pigment fixation occurs when the binder dries out.

The distinction between puro fresco, lime fresco and lime painting techniques may be problematic if microstratigraphy, layer thickness and carbonation surfaces are not clearly identified [23].

As a contribution to answering the aforementioned questions, we conducted archaeometric analyses of Daunian graves with painted figurative decoration in order to collect data on the techniques, pigments and binders used and therefore contribute to the characterization of the technological know-how of the craftsmen and artists. It was chosen to focus the research on Daunia, as this region—and particularly in the two centres of Arpi and Canosa di Puglia—hosts the largest number of painted graves with figurative scenes. Furthermore, also found in this region are polychrome painted pots and wall paintings in the domestic sphere, two artistic domains that from a technological point of view are potentially linked to tomb painting [8,24,25]. Figurative paintings are preserved or described for eleven Daunian funerary monuments; in one additional case a painted relief is attested. Of these monuments, in two cases the friezes are no longer preserved (Ipogeo Lagrasta 1 in Canosa di Puglia, Ipogeo Cassano in Salapia) and two others have already undergone archaeometric analysis (Ipogeo della Medusa and Tomba del Trono in Arpi; [15,26]). The present contribution therefore introduces analyses for five additional Daunian tombs, two in Arpi (Tomba della Nike, Tomba dei Cavalieri) and three in Canosa di Puglia (Ipogeo del Cerbero, Ipogeo Scocchera B, Ipogeo Sant’Aloia). The present analyses thus promise a considerable broadening of our knowledge concerning the techniques and materials used in the figurative decoration of graves in Hellenistic Daunia.

## 2. Results

### 2.1. Microscopy

In this section are reported the results of the POM and SEM-EDS analyses, which focused on the stratigraphy, the composition of the binders used in painted plasters and the relation between the finishing layer and the painting film. A synthetic view of the petrographic features is shown in Table 1.

#### 2.1.1. Tomba della Nike

The sample of Tomba della Nike (sample 3) consists of two layers of lime mortar (A and B) and a painting layer (P) (Figure 1). The preparatory layer A is visible to a thickness of about 800 µm and has an arenaceous texture, over which a finishing layer (B) is applied (600 µm). The painting layer of about 30 µm is optically dense and brownish.

The aggregate of layer A is about 15% vol. and consists of calcareous sand, ferruginous lumps and rare quartz grains. The binder is of varying sizes from micro- to crypto-crystalline. A large crystal of spathic calcite is visible. The presence of pores indicates the incorporation of air during mixing (primary porosity). Layer B is quite compact and presents a fine texture. Smoothing of layer B is also indicated by the alignment of primary pores to the surface.

The absence of lime nodules points to the ageing of lime putty. Visible but not sharp contact between layers A and B indicates that layer B was applied over the still fresh layer A.

EDS maps show that layer A was obtained by mixing lime putty and some clayey sediment to obtain a mildly hydraulicised mortar. Layer B is richer in pure lime and shows carbonation on the surface which includes the painting film (puro fresco).

#### 2.1.2. Tomba dei Cavalieri

The sample of the Tomba dei Cavalieri (sample 12) is layered, with a single layer of lime plaster (A) and two painting layers (P1 and P2) (Figure 1).

Fine calcareous sand occurs as aggregate and some ferruginous lumps. As for the binder, several calcination relics were observed under the microscope. Its texture ranges from micro- to crypto-crystalline. The primary pores prevail over the secondary ones. The thin brown film (P1) has a merging boundary with layer A, also highlighted by the hued band at the top of the layer. The painting layer P2 is 60 µm thick and is partially detached from P1.

SEM-EDS analysis showed that the layer A is a mixture of lime putty and some clayey sediment (Figure 2). The film P1 is red puro fresco (goethite) applied on layer A. Layer P2 is a painted layer of about 100 µm thick on film P1 and shows the following mean chemical composition: CaO = 35% wt.; SO3 = 25% wt.; Al_2_O_3_ = 25% wt.; SiO_2_ = 8% wt.; K2O = 4% wt., which points to a mixture of lime and K-alum with charcoal.

#### 2.1.3. Ipogeo Scocchera B

The sample of Ipogeo Scocchera B (sample 23) has three layers: a preparatory one (A) and two lime paints (P_1_ and P_2_) (Figure 1). Layer A has a fine sand texture, with an aggregate composed of carbonate grains and bioclasts, ferruginous lumps and rare silicates. Layer P_1_ is a lime paint 350 µm thick. Typical drying fissurations occur. The boundary between layers A and P_1_ is irregular and sharp. The lime paints were applied on layer A dry. EDS analysis on lime slaking nodules reveal that pure lime was used as binder (CaO: 93 ÷ 97% wt.).

Sample 24B consists of four layers: a lime-rich layer (A), a thick, roughly prepared layer (B), a finishing layer (C) and a lime paint (P_1_). Layer A is optically dense and almost free of aggregate (<5% vol.). Setting fissuration is almost perpendicular to the external surface. The texture of the binder is cryptocrystalline. No lime lumps were observed. Layer B is 4 mm thick and exhibits a nodular texture, suggesting incomplete slaking. A few fragments of spathic calcite occur as aggregate (<5% vol.). Primary rounded porosity and setting fissures are diffused in the layer (15% vol.).

A careful analysis shows that the partial detachment between layers A and B was infilled by an agglomeration of dust and debris deposited over time.

EDS analyses on sample 24B showed that the two layers contain pure lime as binder, as well as layer P1. Gypsum occurs on the outer surface of the painting.

#### 2.1.4. Ipogeo del Cerbero

A lime plaster layer (A) and a painting film (P) were identified in sample 27 from Ipogeo del Cerbero (Figure 1). The texture of the binder is cryptocrystalline and few lime lumps are present. About 15% vol. of aggregate (carbonate sand, spathic calcite and ferruginous lumps) with seriate texture is distributed in the layer. The layer is quite compact, with primary porosity < 5% vol. The paint film P (10 µm) is optically dense and has a clear boundary with the layer A. Layer A is a mixture of lime putty and some clayey sediment. Paint P is a mixture of lime, clay minerals and iron oxides. Some gypsum was detected on the surface (Figure 3).

#### 2.1.5. Ipogeo Sant’Aloia

In sample 38, a nodular texture characterises layer A. The aggregate (10% vol.) is composed of calcareous and quartz sand and ferruginous lumps (Figure 1 and Figure 2). Primary rounded pores and setting fissures were observed (10% vol.). A paint film (P) with variable thickness (ca. 10 µm) was detected on the surface.

A BSD image of the outer surface of layer A shows that lime carbonated before the application of paint film P.

### 2.2. Raman and FT-IR

The results of the Raman investigations of the analysed samples are reported in Table 2. In all samples, in addition to the pigment/colourant responsible for the colour, a peak at about 1085 cm^−1^, characteristic of calcite, was highlighted [7]. Raman peaks assignable to gypsum and/or anhydrite [27] were, instead, highlighted only on the samples from the Ipogeo Scocchera B and rarely seen on a few samples of the Ipogeo del Cerbero, indicating a good state of conservation of the paintings. The presence of gypsum is certainly due to decay.

#### Tomba della Nike

The Raman spectra for the differently coloured areas of pediment and wall band decorated with spirals at Tomba della Nike, except for pink one, are shown in Figure 4.

The pigment used in the blue areas is, as expected, Egyptian blue (sample 2), the old-est (4th millennium BCE) recognized synthetic pigment [28,29,30,31,32] and the most common blue pigment within the Mediterranean region all through the Hellenistic period. It is a synthetic high-fire polycrystalline material consisting of a main crystalline phase of cop-per calcium tetrasilicate, unreacted quartz and glass.

The identification of two pigments with different hues and market values in the red areas (hematite [33,35] and cinnabar [34,39]) indicates special attention paid by the paint-er to the choice of pigment. Red ochre was used for the spirals (sample 7), while cinnabar (HgS) for the drops of blood on the leg of the unhorsed and wounded warrior (sample 4). Cinnabar was very expensive compared to the natural red earths. Greek and Roman sources report its rarity and preciousness (Theophrastos, De Lapidibus 58; Vitruvius, De Architectura VII, 8–9; Pliny, HN XXXIII, 111 and 118). Therefore, it was used in paintings, and/or parts thereof, of important economic/symbolic value.

As for the other coloured areas, pigments typical of the Mediterranean area in the Hellenistic period were identified: calcite (and thus lime white) for white, goethite (α-Fe^3+^O(OH)) for yellow and hematite (Fe_2_O_3_) for brown as well as red.

Regarding the pink pigment (sample 1), the large fluorescence background makes the detection of any Raman signal impossible and suggests the presence of an organic dye. The absence of Br, which allows the presence of Tyrian purple to be excluded, as well as the presence of a strong pink-orange UV fluorescence (Figure 5), suggest the use of madder lake [40], an organic dye from the roots of Rubiacee plants.

According to ancient recipes, the colourant components (and their precursors) are extracted from madder root into an aqueous solution and then adsorbed or complexed with an inorganic substrate to generate an insoluble substance suited for painting [41,42,43,44]. Chalk, limestone or shell, alum, white clays or earths have been utilized since antiquity [42,44,45,46].

Despite the fact that the substrate determines the color, transparency, working capabilities and durability of the pigment, only a few studies devoted to substrate analysis can be found in the literature [41,44].

Between the madder lake pigment layer—thickness about 20 microns—and the plaster, a Ca-based layer is present (Figure 6).

The composition and distribution of Ca, S, C and O inside the various layers identified the use of lime for the creation of this layer, probably applied to enhance the pink colour.

As for the pink paint layer, X-ray maps (Figure 7) and EDS analyses (Table 3) show a Ca-poor outer domain, characterised by the homogeneous distribution of Al, Si, K and S, pointing to the deliberate addition of potash alum (Al_2_(SO_4_)_3_K_2_SO_4_·12H_2_O) for the preparation of pigments and clay as filler [44,45,46,47].

The SERS spectrum of the paint layer (Figure 8) results in intense bands in the range 1250–1500 cm^−1^, in particular at 1280 and 1325 cm^−1^ attributed by Doherty et al. [48] to the ν(CO)/ν(CC)/δ(CCC) and the ν(CC), and the band at 1472 cm^−1^ attributed to the ν(CO)/ν(CC)/δ(CH) of both alizarin and purpurin [36,48,49].

With respect to the black pigment, the shape of the Raman peaks is consistent with an amorphous carbon-based pigment (Figure 7). Within the great variety of carbon-based pigments—crystalline carbons, flame carbons, charcoals and black earths—it is possible, as suggested by some authors [37,38,50], on the basis of the parameters obtainable from the Raman peaks of the D and G bands, to trace the starting material used. Based on the position of bands (at approximately 1350 cm^−1^ and 1591 cm^−1^ respectively) and width (about 250 cm^−1^ and 85 cm^−1^ respectively), the raw material used for the black pigment of both the Ipogeo Scocchera B and the Ipogeo del Cerbero can be attributed to charcoal.

Due to the extremely limited sampling quantities, FTIR analyses were carried out on representative samples of tombs for which microscopic analysis or the presence of an organic dye (such as in the case of sample 1 from the Nike tomb) might lead one to suspect the presence of an organic binder. Samples 1, 15 and 23 were analysed. The spectra obtained did not reveal the presence of binders in any of the analysed samples (Figure 9).

In the spectrum of sample 15, in addition to calcite, signals attributable to ethyl silicate, a consolidant frequently used in the restoration of frescoes, are visible in the 1050–1100 area.

## 3. Experimental

### 3.1. Samples

Wall paintings from two tombs in Arpi (Foggia) and three in Canosa di Puglia were analysed (Figure 10 and Figure 11 and Table 4).

### 3.2. Techniques

Non-destructive portable techniques are the ideal choice for gathering information on-site and preserving the integrity of the artwork when wall paintings need to be analysed, mainly in terms of the examination of pigments/colourants.

However, the presence on the surface of compounds, such as acrylic polymers, organic-silicon, lime and gypsum, which are frequently employed on wall paintings for protection/consolidation, limits the application of non-destructive portable instruments (without counting complications related to sensitivity and lateral resolution, frequently not sufficient to obtain information about all layers).

Although on the wall paintings analysed in this study there are no records of restorations carried out, and because of the need to also analyse mortar/plasters, micro-sampling (few millimetres sized, sampled along existing fractures) was preferred in this project and analyses were performed on the cross-section of the sampled fragment.

#### 3.2.1. Spectroscopic Investigations

##### Raman and SERS

Raman investigations were carried out employing a LabRAMHR Evolution^®^ (Horiba^®^, Kyoto, Japan) spectrometer equipped with a Peltier-cooled charge-coupled device detector (CCD), He-Ne (633 nm) and Ar (488–514 nm) lasers, a BH2^®^ microscope (Olympus Corporation^®^, Tokyo, Japan) and an Ultra-low wavenumber module, which allows Raman spectroscopic information in the sub-100 cm^−1^ region. Four objectives, 10X, 50X, 100X (Leica DMLM microscope) and long working distance 50X (Olympus (Japan), were used to focus the laser beam on the samples. The system has a spatial resolution of 1 μm and a spectral resolution of about 1 cm^−1^. The instrument was calibrated with a Si (111) standard (520.5 cm^−1^). A linear baseline was subtracted from the raw spectra using the software LabSpec6^®^ (Horiba^®^, Kyoto, Japan). Substances were identified by the comparison with reference spectra.

The SERS technique was used to identify the pink colorant from the Tomba della Nike, for which, due to the fluorescence associated, no Raman spectra could be obtained. Silver colloids were used as SERS signal amplifiers. The Ag colloids were synthesised following the procedure of Lee and Meisel [53]. Deionised water was added to 0.09 g silver nitrate and the solution was brought to the boil under vigorous magnetic stirring. 10 mL of 1% trisodium citrate was added and the solution boiled for thirty minutes. After cooling to room temperature, 1 mL aliquots of the Ag solution were centrifuged for 15 min to concentrate the silver nanoparticles. Ten successive centrifugation cycles were performed, discarding the supernatant each time and adding an additional millilitre of the solution containing the silver colloids. The colloidal paste thus obtained was applied to the sample prior to Raman analysis.

##### FT-IR Investigations

Attenuated total reflection spectra were collected using FT-IR Perkin Elmer Spectrum Two equipped with a ZnSe crystal. Spectra (64 scans) were acquired on the wavenumber range of 400–4000 cm^−1^, with an optical resolution of 2 cm^−1^. Spectrum software was used for spectral collection and data manipulation.

#### 3.2.2. Microscopic Investigations

##### POM and SEM-EDS Investigations

Petrographic observations were carried out on thin sections through a “ZEISS Axioskop 40 POL” transmitted and reflected light (VIS-UV) polarizing optical microscope (POM). An EXFO X-Cite 120 external VIS-UV illumination system was used for observations in reflected light. Images were acquired with a Nikon DS-Fi1c CCD camera with associated Nikon Digital Sight DS-U2 controller unit. The abundance of the aggregate and macro-porosity was obtained by visual estimation using comparison charts [54]. A ZEISS/LEO EVO 50XVP scanning electron microscope (SEM) was operated at 15 kV and 500 pA probe current, with an 8.5 mm working distance. Thin sections were coated with graphite for SEM investigations. Energy dispersive spectrometric (EDS) microanalyses and X-ray maps were obtained using an X-MaxN (80 mm^2^) SDD detector and AZtec software (Oxford Instruments). About 25,000 cps output as average count rate on the whole spectrum and a counting time of 50 s for microanalysis were considered. The correction of X-ray intensity was performed following Pouchou and Pichoir [55]. Different Micro-Analysis Consultants Ltd. (St Ives, Cambridgeshire, UK) mineral standards were used to check the accuracy of the analytical data. Relative errors ((|measured composition − certified composition|/certified composition) × 100%) of four of the standards used for element calibrations (augite, almandine, pyrope and orthoclase) are reported in Table 5.

POM and SEM analyses were performed on a selection of samples (3, 12, 23, 24B, 27, 38) in which it was possible to recognize the plaster stratigraphy.

Table 6 summarizes the analytical techniques used in the analysis of each sample.

## 4. Discussion and Conclusions

The study conducted involved the analysis of 35 samples of painted plaster taken from five Hellenistic graves in Daunia, northern Apulia, and was carried out in order to identify the pigments and binders used and to understand the painting technology involved. Stratigraphic analysis by optical and electron microscopy observation made it possible to define the structural characteristics of the various layers, while Raman and SERS investigations were used to identify the pigments. The results allow us to operate a differentiation in both technique and materials used between a group of four monuments analysed and one grave, the Tomba della Nike, which differs from the others in many respects. Although all plaster layers have lime as binder, the sample from the Tomba della Nike shows a higher quality in terms of lime hydration and plaster homogeneity compared to the other samples. The diffuse presence of hydration nodules and sometimes of calcination relics (sample 12) occurs in the remaining samples. This evidence suggests that all layers were prepared with lime putty, although maturation probably occurred only for the sample 3. The aggregate content is generally low (<15%) and fine-grained due to the thinness of the plaster layers. The aggregate consists essentially of calcareous sand and ferruginous lumps as an accessory component. The boundary between the plaster and paint layers in the analysed samples points essentially to fresco application in the Arpi tombs and fresco combined with secco lime painting in Canosa di Puglia [23]. This suggests a plurality of technological knowledge in the workshops that contributed to the decoration of these graves, or even to competition between different specialized workshops. The presence of K-alum in sample 1 (Tomba della Nike) in the madder lake preparation and mixed with lime and charcoal in a secco layer of sample 12 (Tomba dei Cavalieri) seems to be another feature of Arpi that should be verified in further investigations.

In four of the five tombs analysed (Tomba dei Cavalieri, Ipogeo del Cerbero, Ipogeo Scocchera B and Ipogeo Sant’Aloia) the application of the pictorial layer directly onto a more or less coarse preparatory layer was observed, with an intermediate layer, aimed at smoothing the plaster before the application of the pictorial layer, seeming to be absent. Regarding the pigments used, lime for white, charcoal for black and hematite for red were identified in all four tombs, goethite for yellow in three cases and Egyptian blue for blue in two. In the case of the Tomba dei Cavalieri, the red ochre and Egyptian blue pigments identified by Brecoulaki [26] on the dress of the female figure on the right slab of the chamber were also identified through the new analyses. The pigments fit into the typical palette used in Hellenistic tomb painting both in Apulia and more generally in Southern Italy, Etruria and Macedonia. A palette composed of these four pigments—or, in some cases, only three or two of them—is the one most frequently detected in analyses of late-classical and Hellenistic tombs both in indigenous centres in Apulia [26,56,57,58,59,60], in Taranto [61,62], in Paestum [26], in Etruria [63,64,65,66] and in Macedonia [67]. These were therefore widespread pigments, probably locally extracted or produced, the processing of which was practiced on a large scale. In Etruria and, in isolated cases, Southern Italy, the use of a palette composed of lime for white, charcoal for black, hematite for red and Egyptian blue for blue is documented as early as the 6th and 5th centuries BCE [26,59,68]. Finally, the parallelism with the execution technique of the wall paintings of the Domus del Mosaico dei Leoni e delle Pantere in Arpi should be stressed. Here, the same four pigments, as well as a green earth, are applied with fresco and secco techniques on two preparatory layers [8].

The fifth monument, the Tomba della Nike, differs from the previous ones in both the technique adopted and the materials used. Stratigraphic analyses revealed the presence of three layers: a preparatory layer, a compact finishing layer with fine texture and, finally, the pictorial layer. In contrast to what was observed in the other tombs, the workmanship is particularly refined, with slaked lime consisting of well-selected material (fragments of fine and homogeneous grain size) and well worked (absence of calcination relics, parallel alignment of pores to the surface).

With respect to pigments, in addition to those already found in the other graves, namely lime for white, goethite for yellow, hematite for red and Egyptian blue for blue, analysis revealed the presence of cinnabar and madder lake. Cinnabar was used for the deep red blood of the wounded warrior in the centre of the scene. As mentioned, cinnabar is referred to by ancient sources as quite expensive and held in high estimation (Pliny, HN XXXIII, 111 and 118). According to Theophrastus (4th century BCE), the ore was mined in Colchis, Ephesus and Iberia (De Lapidibus 58–59; Pliny HN XXXIII, 113–114), although it is not clear whether the author refers to Spain or rather to Transcaucasian Iberia [44]. At the beginning of the imperial period, Spain is mentioned as the main area of origin of cinnabar (Vitruvius, De Architectura VII, 8–9; Pliny HN XXXIII, 118). The pigment was very sensitive to light and it was preferable to use it in interiors (Vitruvius, De Architectura VII, 9; Pliny HN XXXIII, 122). The use of cinnabar is documented in pre-Roman tomb painting from the 6th century BCE onward in Etruria and in the 5th century BCE in Lucania and Peucezia [26,68]. In the 4th century BCE there are numerous attestations of the use of cinnabar either to emphasise certain details with strong tones or associated with other pigments to create mixed shades such as pink or purple. Cinnabar was detected especially in Macedonia [67], but also in some Etruscan and Tarentine tombs [13,26,69]. At Arpi, in addition to the Tomba della Nike, cinnabar is used in two compositions on the pediment of the Tomba del Trono: mixed with Egyptian blue, charcoal black and ochre to obtain a blue colour, and mixed with organic lake extracted from madder roots (identified as /Rubia peregrina/), ochre and charcoal black to obtain a dark pink [26].

For the pink background, the analyses identified the presence of an organic lake: the colouring components were extracted from the madder root with K-alum and mixed with clay to generate an insoluble substance suitable for painting [44], applied without the use of organic binders on a thin layer of lime. The lake is extracted from the roots of the madder, a herbaceous perennial plant originating in western Asia but cultivated in southern Europe, and has a colour that can vary from scarlet to violet-pink. Its use is documented for dyeing textiles and for application on dry substrates [40,70].

According to Pliny, the plant was widespread in Italy and the provinces and used to dye wool and leather (Pliny, HN XIX, 47). This type of lake is documented in other tombs from Arpi. On the tympanum of the Tomba del Trono, it was mixed with particles of cinnabar, red ochre and carbon black and applied dry with organic binder for some details of the throne [26]. It has also been identified on a painted figured capital from the Tomba della Medusa [15].

In addition to Daunia, its use in 4th-century tomb painting is also documented in Campania, Lucania and especially Macedonia [26,40,67].

There is also evidence of its use for the colouring of the background of polychrome pots from Arpi and Centuripe (Sicily), where the lake is applied dry on a layer of kaolin [14,15,71].

The Tomba della Nike is thus distinguished by a refined technique and the use of imported, delicate pigments such as cinnabar or those requiring special handling such as madder lake. In this tomb the experience gained in polychrome pottery painting seems to have played an important role: the pink background, the use of madder lake and, last but not least, the iconography and style of the figurative scene link this tomb to the polychrome tempera painting on ceramics widespread in Arpi and Canosa di Puglia [16,24]. At least in the case of the Tomba della Nike it is thus possible to postulate that a specialised Arpanian workshop produced both painted ceramics and wall paintings [72,73] using the same techniques and materials.

## Figures and Tables

**Figure 1 molecules-28-01055-f001:**
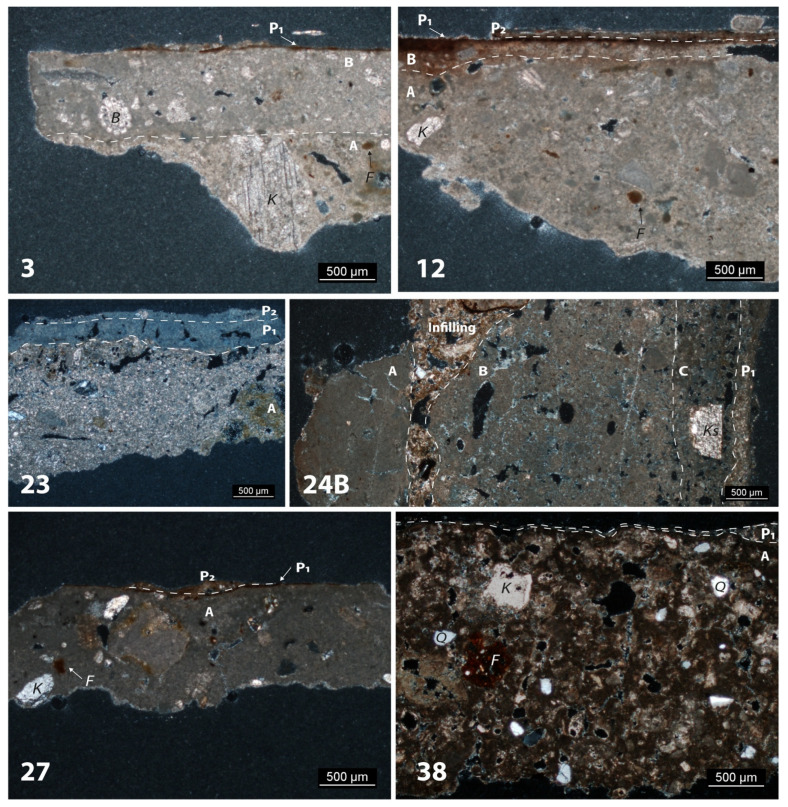
Photomicrographs (2.5X; XP) of samples (white bold numbers) of the different tombs. Plaster (A, etc) and painting (P_1_, etc) layers are indicated, with their relative stratigraphic boundaries (dasched lines).

**Figure 2 molecules-28-01055-f002:**
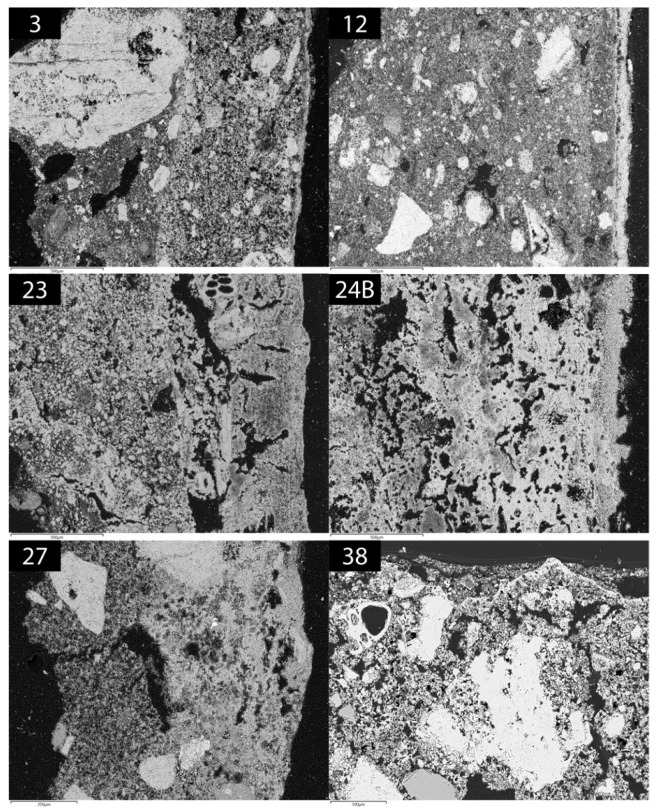
SEM-BSE photomicrographs of thin sections of analysed samples.

**Figure 3 molecules-28-01055-f003:**
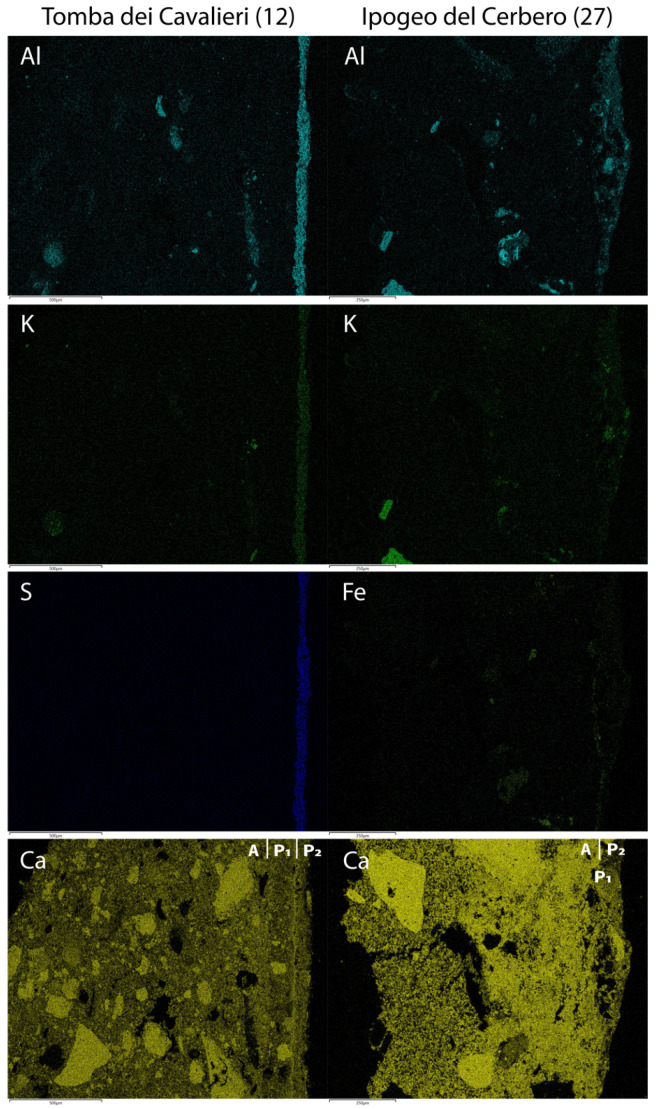
X-ray chemical maps showing and relative stratigraphy of samples 12 and 27. See Table 1 for details.

**Figure 4 molecules-28-01055-f004:**
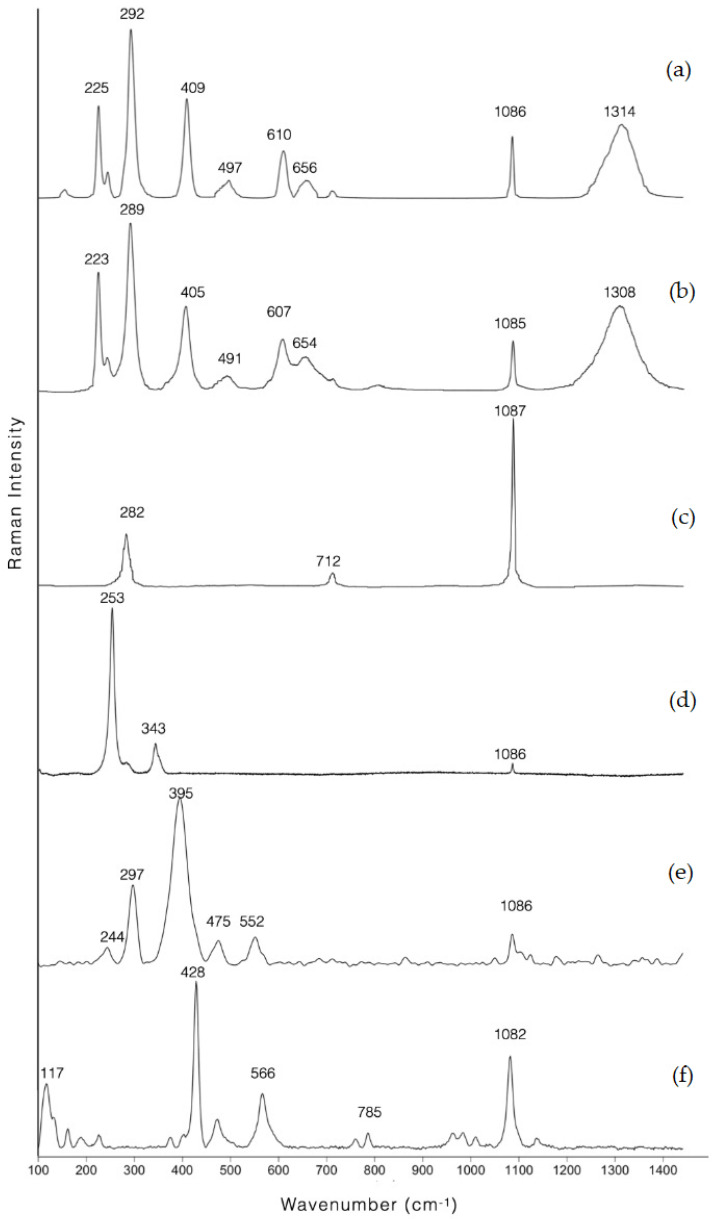
Representative baseline−subtracted Raman spectra of the samples of the tomb of Nike; (**a**) sample 7: red, (**b**) sample 6: brown, (**c**) sample 5: white, (**d**) sample 4: red, (**e**) sample 3: yellow, (**f**) sample 2: blue. Acquisition parameters: 514 nm laser for sample 2 and 633 nm one for all the other samples, 50X LWD objective, 0.6 mW, 3600 s.

**Figure 5 molecules-28-01055-f005:**
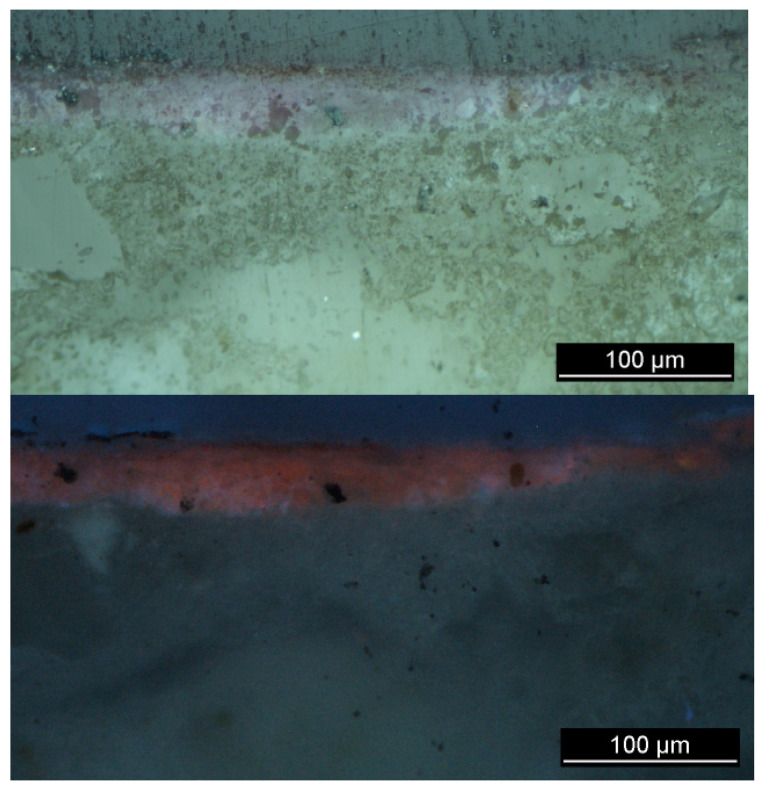
OM photomicrographs of cross sections under white (**above**) and UV-light (**below**) illumination of sample 1 (pink) of the tomba della Nike showing the strong pink-orange-coloured luminescence typically seen under ultraviolet wavelength for lake pigments.

**Figure 6 molecules-28-01055-f006:**
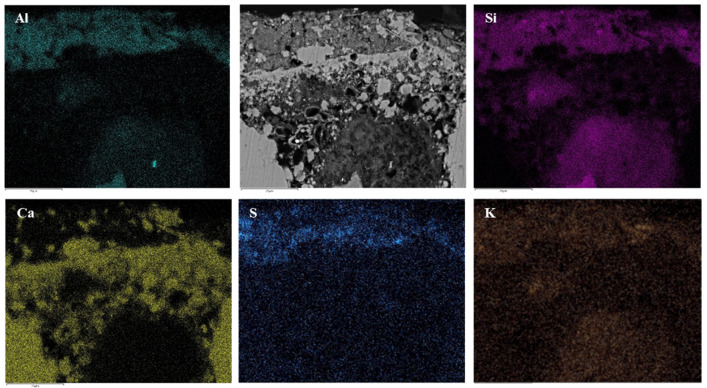
SEM-BSE photomicrographs and X-ray maps (Al, Si, K, S, Ca) of cross section of sample 1 of the tomba della Nike showing, from upper to lower, madder lake pigment, Ca-based layer and plaster.

**Figure 7 molecules-28-01055-f007:**
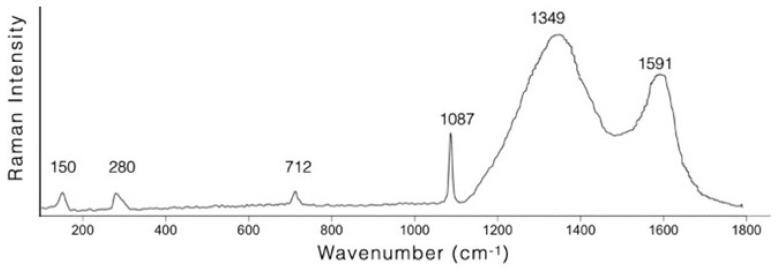
Representative baseline-subtracted Raman spectrum of sample 26. Acquisition parameters: 633 nm laser, 50X LWD objective, 0.6 mW, 3600 s.

**Figure 8 molecules-28-01055-f008:**
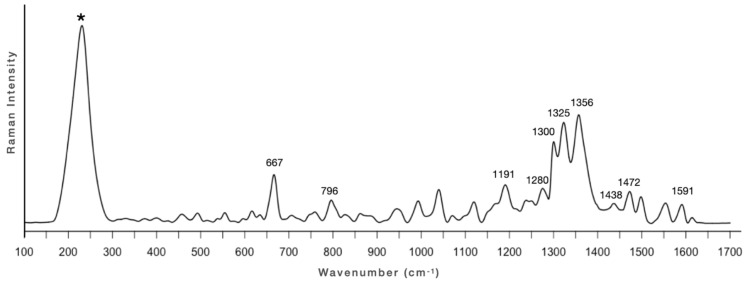
SERS spectrum of sample 1 of the tomba della Nike. Acquisition parameters: 514 nm, 50X LWD objective, 0.4 mW, 3600 s. Peak arising from the silver colloidal paste is labelled with an asterisk. Attribution of peaks based on reference [47].

**Figure 9 molecules-28-01055-f009:**
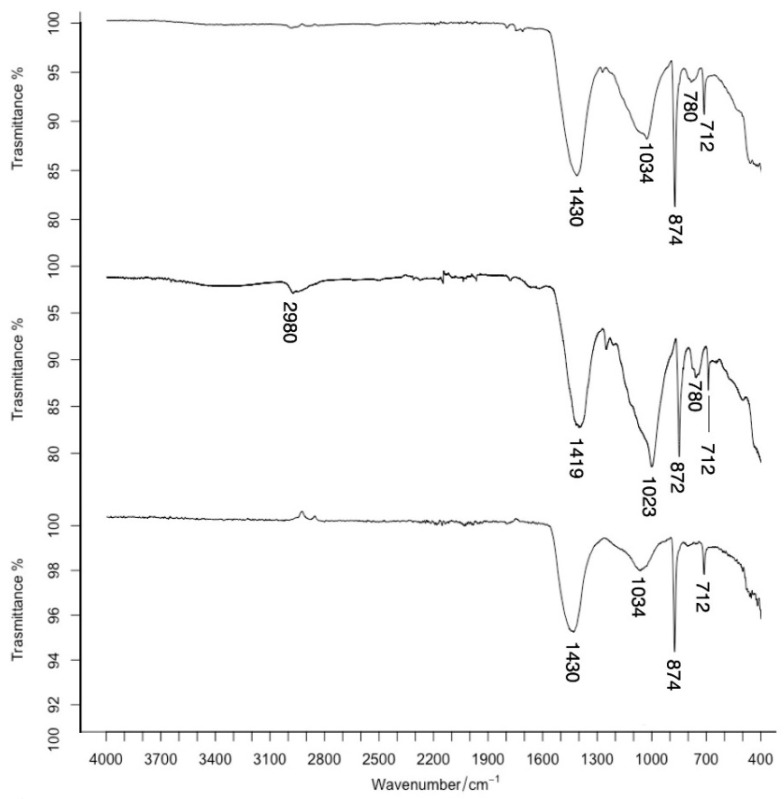
FT-IR spectra of samples 1, 15 and 23 (from top to bottom).

**Figure 10 molecules-28-01055-f010:**
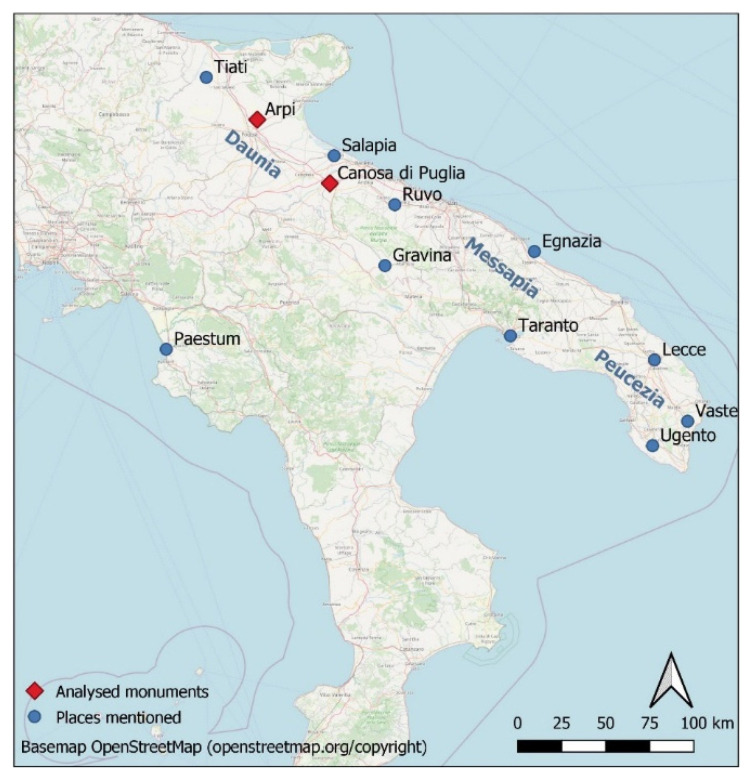
Places mentioned in the text.

**Figure 11 molecules-28-01055-f011:**
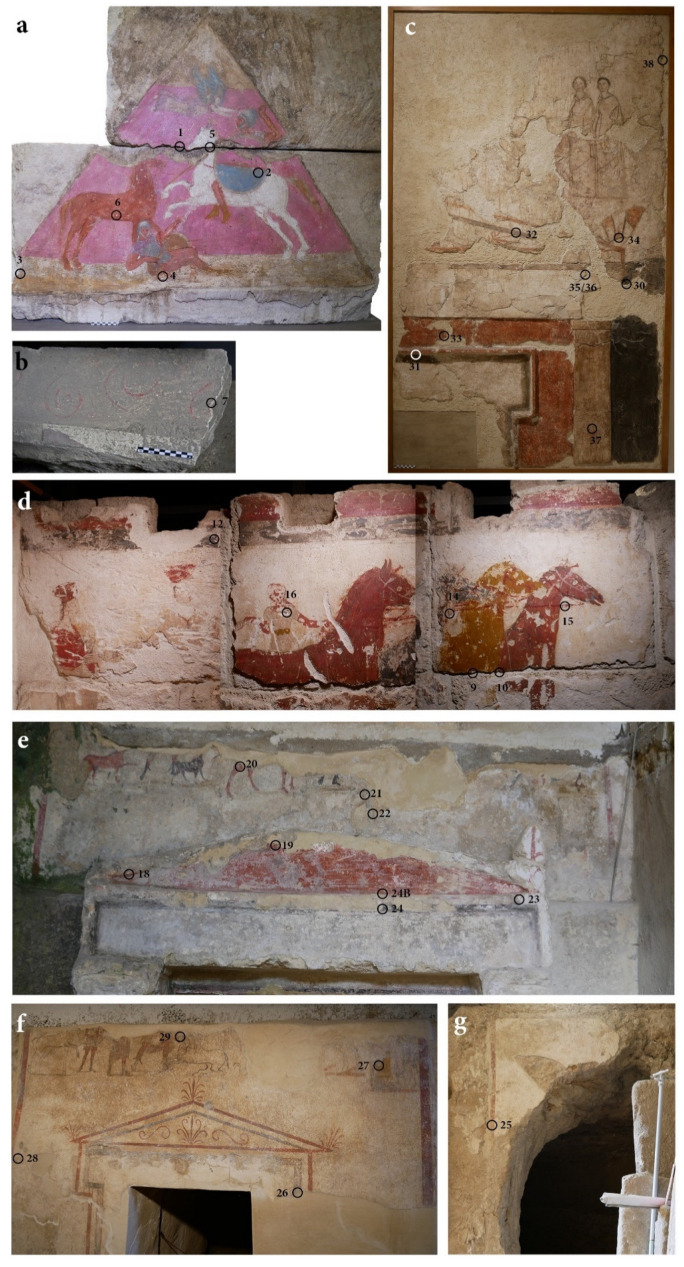
Sampling areas: Tomba della Nike pediment (**a**), wall band with spirals (**b**); Ipogeo Sant’Aloia (**c**); Tomba dei Cavalieri (**d**); Ipogeo Scocchera B (**e**); Ipogeo del Cerbero (**f**,**g**). Images used by permission of Soprintendenza Archeologia, Belle Arti e Paesaggio per le Province di BAT e FG (**a**,**b**,**d**–**g**) and Direzione Regionale Musei Puglia-Ministero della Cultura (**c**).

**Table 1 molecules-28-01055-t001:** Petrographic features of the lime plasters and paintings after POM and SEM-EDS analyses. Abbreviations: H = homogeneous; N = nodular; K = calcareous sand; Ks = spathic calcite; B = bioclasts; F = ferruginous lumps; Q = quartz; Cm = clay minerals. The layers are named from the inside out. The boundary type refers to the transition with the next layer.

	Sample	Layer	Boundary	Binder Structure	Aggregate or Pigment	Painting
Tomba della Nike	3	A	visible	H	K, B, F, Q, Cm	/
		B	merging	H	K	/
		P_1_	/	H	goethite, clay minerals	fresco
Tomba dei Cavalieri	12	A	merging	N	K, B, F	/
		P_1_	sharp	H	goethite	lime fresco
		P_2_	/	H	charcoal, K-alum	lime paint
Ipogeo Scocchera B	23	A	sharp	N	K, B, F, Q	/
		P_1_	merging	H	lime	lime paint
		P_2_	/	H	lime	lime fresco
	24B	A	sharp	N	K	/
		B	merging	N	K, Ks	/
		C	merging	N	K, Ks	/
		P_1_	/	H	charcoal	lime fresco
Ipogeo del Cerbero	27	A	merging	N	K, Ks, B, F	/
		P_1_	visible	N	goethite	fresco
		P_2_	/	H	lime, clay minerals	lime paint
Ipogeo Sant’Aloia	38	A	sharp	N	K, Q, F	/
		P_1_	sharp	H	charcoal	lime paint

**Table 2 molecules-28-01055-t002:** Raman bands and highlighted pigments of all the analysed samples.

Sample	Raman Bands (cm^−1^)	Pigment/Colourant Detected	References
5, 16, 23, 28, 35, 36—white	282, 712, 1085	Calcite	[7]
2, 14, 22—blue	117, 137, 162, 226, 375, 428, 474, 566, 760, 785, 983, 1010, 1082, 1137	Egyptian blue	[28,29,30,31,32]
3, 9, 21, 27—yellow	242, 299, 393, 478, 552	Goethite	[33]
4—red	252, 287, 346	Cinnabar	[34]
6, 37—brown	225, 242, 292,409, 497, 610, 1310	Hematite	[33,35]
7, 10, 15, 18, 19, 20, 24B, 25, 32, 33—red	225, 242, 292,409, 497, 610, 1308	Hematite	[33,35]
1—pink	667, 796, 1191, 1280, 1300, 1325, 1356, 1438, 1472, 1591	Madder lake	[36]
24, 26, 29, 30, 31, 34—black	1350, 1591	Charcoal	[37,38]

**Table 3 molecules-28-01055-t003:** EDS microanalysis (*w*/*w*%) of pigment layer and the Ca-based layer of sample 1.

Sample	Na_2_O	MgO	Al_2_O_3_	SiO_2_	P_2_O_5_	SO_3_	K_2_O	CaO	FeO	Cl
pigment layer	0.38 ± 0.04	0.58 ± 0.04	23 ± 0.37	53.21 ± 0.26	0.62 ± 0.07	8.88 ± 0.12	3.52 ± 0.06	6.96 ± 0.08	1.25 ± 0.1	0.88 ± 0.06
Ca-based layer	-	0.61 ± 0.06	0.97 ± 0.26	5.2 ± 0.1	-	0.57 ± 0.09	0.45 ± 0.06	90.99 ± 0.32	0.37 ± 0.12	0.28 ± 0.04

**Table 4 molecules-28-01055-t004:** Studied tombs.

	Tomb	Dating	Typology	Discovery	Figurative Paintings	Reference
Arpi	Tomba della Nike	Late 4th to early 3rd century BCE	Underground chamber tomb: uncovered dromos and vestibule; barrel-vaulted chamber	2003	Façade of the chamber: Pediment: figurative scene on a pink background (a knight on a white horse is crowned by a flying Nike while wounding a brown horse with a spear; the defeated and wounded opponent is lying on the ground); gable’s slopes: red-coloured spirals and flowers	[16,51]
	Tomba dei Cavalieri	Second half of the 4th century BCE	Semi-chamber tomb built of stone slabs: vestibule and rectangular chamber	1982	Walls of the vestibule: red and black horizontal bands; walls of the chamber: two horizontal bands, continuous figurative frieze depicting a funeral procession (from right to left: quadriga led by a charioteer, two female figures one of which—the deceased—is probably in the act of climbing on the quadriga, two armed horsemen, a chariot with an armed male figure), remains of a second, unrecognisable figurative frieze	[26,51]
Canosa di Puglia	Ipogeo Scocchera B	First half of the 3rd century BCE; the tomb was used for about a century	Underground chamber tomb: dromos with two side chambers, corridor, central room, three burial chambers	1895; 1979 (rediscovery)	Main entrance at the end of the dromos: naiskos (relief and painted architectural elements); above the pediment: lower part of a figurative scene possibly depicting the journey to the afterlife (horses, human figures, pytomorph elements)	[14,52]
	Ipogeo del Cerbero	Second half of the 4th century BCE (first phase); early 3rd century BCE (additional chamber with paintings)	Underground chamber tomb: dromos, central rectangular vestibule, three burial chambers	1972	Entrance to the additional chamber: painted as Doric doorway with pediment and acroteria (black and red strips); above the pediment: figurative scene representing journey the to the afterlife (Cerberus, Hermes Psychopompos, the deceased, a warrior with a horse, a horse and two female figures)	[14]
	Ipogeo Sant’Aloia	3rd century BCE	Underground chamber tomb: dromos, central room, three burial chambers	1956	Façade of the chamber: naiskos (relief and painted architectural element); pediment: palm tree and the tail of a sea creature; above the pediment: figurative scene (horseman, two female figures, back of horseman)	[14]

**Table 5 molecules-28-01055-t005:** SEM-EDS precision and accuracy. Mean of 10 microanalyses on four reference standards (augite, almandine, pyrope and orthoclase) with the relative standard deviation (σ_rel_), certified values from Micro-Analysis Consultants Ltd. (U.K.) and relative error (RE_accuracy_). n.d. = not detected.

	Augite				Almandine				Pyrope				Orthoclase			
	Mean	s_rel_	Certified Value	RE_accuracy_	Mean	s_rel_	Certified Value	RE_accuracy_	Mean	s_rel_	Certified Value	RE_accuracy_	Mean	s_rel_	Certified Value	RE_accuracy_
SiO_2_	46.2	0.09	45.93	0.59	41.25	0.13	41.59	0.82	42.22	0.1	41.71	1.22	64.1	0.06	64.3	0.31
TiO_2_	2.39	0.05	2.29	4.37	0.57	0.04	0.52	9.62	0.59	0.05	0.54	9.26				
Al_2_O_3_	8.75	0.04	9.02	2.99	22.59	0.1	22.66	0.31	21.05	0.06	21.79	3.40	20.03	0.06	19.9	0.65
Cr_2_O_3_	0.14	0.08	0.08	75.00					1.9	0.07	1.74	9.20				
FeO	7.2	0.11	7.42	2.96	12.64	0.12	12.69	0.39	8.88	0.09	8.81	0.79				
MnO	0.06	0.06	0.07	14.29	0.38	0.04	0.22	72.73	0.36	0.04	0.36	0.00				
MgO	12.05	0.03	11.88	1.43	17.36	0.09	16.85	3.03	20.69	0.08	20.29	1.97				
CaO	21.77	0.09	21.62	0.69	5.37	0.05	5.23	2.68	4.43	0.03	4.5	1.56	0.18	0.03	0.24	25.00
Na_2_O	0.85	0.03	0.79	7.59	n.d.	---	0.04						3.86	0.16	3.7	4.32
K_2_O	n.d.	---	0.01		n.d.	---	0.01						11.38	0.23	11.4	0.18
Tot.	99.41		99.11		100.44		99.81		100.12		99.9		99.55		99.54	

**Table 6 molecules-28-01055-t006:** Analytical techniques used for each analysed sample.

Tomb	Sample/Colour	Analysis Technique
Tomba della Nike	1. Pink (background)	Raman; OM (white and UV-light illumination); SERS; SEM-EDS; FT-IR
	2. Blue (shield)	Raman
	3. Yellow (background)	Raman; POM; SEM-EDS
	4. Red (blood)	Raman
	5. White (horse)	Raman
	6. Red/brown (horse)	Raman
	7. Red (spirals)	Raman
Tomba dei Cavalieri	9. Yellow (horse)	Raman
	10. Red (horse)	Raman
	12. Gray-white (leaf)	POM; SEM-EDS
	14. Blue (harness)	Raman
	15. Red (reins)	Raman; FT-IR
	16. White (dress)	Raman
Ipogeo Scocchera B	18. Red (fragments)	Raman
	19. Red (tympanum)	Raman
	20. Red-brown (horse)	Raman
	21. Yellow (ground)	Raman
	22. Blue-light blue	Raman
	23. White (frame)	Raman; POM, SEM-EDS; FT-IR
	24. Black (frame)	Raman
	24-B. Red (fragment)	Raman; POM; SEM-EDS
Ipogeo del Cerbero	25. Red (door frame)	Raman
	26. Black (frame)	Raman
	27. Yellow (robe)	Raman; POM; SEM-EDS
	28. White (background)	Raman
	29. Black (cerberus, outline)	Raman
Ipogeo S. Aloia	30. Black (pillar/side frame)	Raman
	31. Black (inside door frame)	Raman
	32. Red (tympanum)	Raman
	33. Red (door frame)	Raman
	34. Black (acroterion)	Raman
	35. White (architrave)	Raman
	36. White (architrave)	Raman
	37. Brown (pillar)	Raman
	38. White (background)	POM; SEM-EDS

## Data Availability

All data are in the paper.

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
