# Peer review of "Pigments and Techniques of Hellenistic Apulian Tomb Painting"

_molecules, 2023, doi:10.3390/molecules28031055_

Round 1

Reviewer 1 Report

The manuscript of Annarosa Mangone et al. is an interesting multi-analytical study that reports results obtained on several wall painting fragments coming from five different Hellenistic tombs from the Daunia area (Italy). The painting materials and techniques were explored via combined analytical techniques that included SEM-EDS, Raman (including SERS), and FTIR analysis. The introduction provides a comprehensive context for the research work, and the research questions are clearly formulated. Overall the manuscript is well structured and the findings are correlated with previous studies. The manuscript adds to the know-how of painting techniques and materials used within the region. However, there are some issues related to data acquisition and data presentation that the authors need to address and clarify. Please see some comments and suggestions below:

Abstract: I suggest revising the abstract as in its current form it is mainly focused on the obtained data. Include some background/context for the present research as well as some conclusions.

Line 16: When referring to wall paintings I would suggest using “painting technique” instead of “manufacturing technique”. Check throughout the text.

Section 2.1 Samples. It would be great to have a table with all the investigated samples. At this moment is not clear to the reader what type of measurements were carried out on each sample. The table should include a short description of the sample, an image (if possible), and a checklist in terms of methods employed for characterization.

Section 2.2 Techniques. I suggest splitting the methods into subsections. It would be easier to find the technical data/parameters of each particular technique.

Lines 188-190: Were the wall paintings investigated in this study subject to restorations? If so, is there any available information on these past interventions including the materials used for their conservation? If subject to previous restorations, how were the sampling areas selected? This is an important aspect moreover as one of the key questions that the present study addresses are the possible presence of an organic binder. Please add some clarifications.

Lines 219-222, FTIR. Please include the number of scans used.

Section 3.1. Microscopy. Within the subsections (3.1.1 – 3.1.2) the authors discuss results obtained on particular samples. Were the POM and SEM-EDS analyses carried out on selected samples only? If so, on what criteria were the samples selected? Please clarify.

Line 271. “The layer P2 is a secco painting on film P1…”. How did the authors determine at this stage of the study that this layer was a secco? According to Pique and Verri (Organic Materials in Wall Paintings, Getty, 2015), painting a secco means the use of organic binders mixed with pigments and applied to dry plaster. Please clarify. Also, the term a secco should be explained/introduced somewhere in the introduction, so that the reader can have a clear view of the findings.

Line 281: “The limewash (P1) was applied a secco on the layer A”. This sentence is confusing. Please revise.

Figure 2. The authors should include within the various images provided arrow annotations in accordance with the notations used in the text for the various layers and aggregates identified. Same for Figure 3, some arrow annotations would be required.

Figure 4. The caption should be revised. Again, what exactly do the authors mean by secco application?

Table 1. In my opinion, the data provided here are redundant. Instead, a table with a summary of the SEM-EDS results obtained would be more adequate. Please revise.

Lines 329-331. This sentence needs a reference.

Table 2. The caption does not include information on which sample were the data obtained. Please correct.

Table 3. Please include another column were to include the references consulted for the band assignment. Same for Table 4 and Table 5.

Lines 439-441. Why was FTIR analysis carried out only on 3 samples? How were these samples selected? FTIR spectra should be included in the manuscript. Also, based on the fact that the authors inferred that some of the pigments were applied a secco some comments and clarifications should be added within the manuscript on why no organic binder was highlighted within the registered FTIR data.

Lines 461-463. “The boundary between the plaster and painting layers in the analysed samples points to fresco application in the Arpi tombs and secco lime-painting for Canosa”. This sentence needs further clarification. The study cited refers to fresco and lime-paint.

References. In my opinion, the number of self-citations is too high. Yes, the cited papers deal with findings obtained on artifacts from the same region but these are mainly pottery artifacts (References 1 to 16), not wall paintings. I suggest citing the most representative references only.   

Author Response

All suggestions/corrections made by the referee, whom we thank, have been taken on board and the manuscript amended

Reviewer 2 Report

Review of Mangone et al. “Pigments and techniques of Hellenistic Apulian tomb painting”

The authors present detailed results from the chemical analysis of 5 Apulian tomb paintings as part of a larger study of this Hellenistic art form. The research is of interest and provides further important information regarding this understudied art history.  My comments are presented categorically for easier handling by the team of authors.

Scientific Analysis

The quality of the scientific data is high, and the authors are congratulated on generating excellent Raman spectra in particular of real-life samples. Still, there is room for improving the presentation and simplifying the articles many figures to only those that are essential.

After Figure 5, the remaining pigments from the other sites as determined by their Raman spectra are identical to those in Table 3 with the exception of charcoal and madder lake.  Therefore, a lot of article space and distraction could be saved by removing the remaining large figures of repeated Raman spectra and simply use a single table for all artworks to report the results from the other sites. There is no advantage to showing a lot more (nearly identical) spectra of the inorganic pigments that are already well known by spectroscopists.  The one example of plotted spectra is enough to give an idea of the spectral quality. 

Figure 9 says 514nm excitation was used for sample 2, but there is no sample 2 in that figure.  Is this a copy and paste error from Figure 5?

The identification of madder lake is noteworthy and does require its own figure.  In the general description the authors point out that the layer is too fluorescent for normal Raman, shows an orangish visible fluorescence, and contains no Br (to rule out geranium lake??), which are all supportive of an organic component.  Being that the spectrum was acquired using SERS, I would argue that the authors need to overlap the sample spectrum with a reference spectrum of madder lake using the same SERS reagent. As presented in Figure 8, the authors’ SERS spectrum has a very different bandshape and relative peak intensities to the literature cited, even if some of the bands present are consistent between the two experiments (and numerous other bands of similar intensity are different?).  A direct comparison using the same SERS reagent on the same instrument with the same baseline subtraction is the only way to make a convincing argument for the non-hydrolysis identification of madder lake.

Ll 386-390 discussing the EDS maps suggests the presence of Al, Ca, K, and S suggests a clay (why sulfur present?) or alum substrate.  Later, the potential use of alum seems discarded, and clay is mentioned as the substrate (L. 517).  I was not clear why this change of opinion occurred.  To me the use of alum seems more likely based on the S, but then again this could also be due to some gypsum being present.  The authors should clarify the rationale for their identification or else remain uncommitted based on the two possibilities.

Since the authors make special mention of the charcoal black spectrum, it should be presented as its own figure near the discussion of its features that distinguish it from other carbonaceous black pigments.  Line 438:  The authors attribute the carbonaceous pigment to charcoal based on similarities with data presented in research published elsewhere.  It would be useful for them to highlight in more detail the peak ratios. bandwidths, etc calculated from their spectra and the previously reported data that lead to that assignment. For such a nuanced analysis it is otherwise difficult to gauge the reliability of their assignment, especially for an assessment that is so rarely undertaken in our field.

Archaeology

     In the abstract "Daunia area (northern Apulia)". In para 1 "Daunian area (northern Apulia)"

     For the Tomba della Nike (Arpi)--insert the place name the first time, since where specifically it is not spelled out till page 2.

     It says that the madder was applied on lime (line 519) and then it says that lime can’t be applied directly to fresh lime (line 522). One seems inconsistent.

     For the discussions of cinnabar, the excellent review paper is missed Gliozzo. (2021). Pigments — Mercury-based red (cinnabar-vermilion) and white (calomel) and their degradation products. Archaeological and Anthropological Sciences, 13(11). https://doi.org/10.1007/s12520-021-01402-4

     Line 530: Please correct: “Its use does not seem to be known in Etruria…”, for example:

Gagliano Candela, Lombardi, L., Ciccola, A., Serafini, I., Bianco, A., Postorino, P., Pellegrino, L., & Bruno, M. (2019). Deepening Inside the Pictorial Layers of Etruscan Sarcophagus of Hasti Afunei: An Innovative Micro-Sampling Technique for Raman/SERS Analyses. Molecules (Basel, Switzerland), 24(18), 3403–. https://doi.org/10.3390/molecules24183403

     The article by Aceto offers an excellent review of organic pigments in wall paintings and is missed in the present work.  Aceto M. (2021). Pigments—the palette of organic colourants in wall paintings. Archaeological and Anthropological Sciences. https://doi.org/10.1007/s12520-021-01392-3  Which says on the point of madder in Etruria, "Brecoulaki (2014) reports that madder lake was frequently used with Egyptian blue in paintings from the Bronze Age to Hellenistic times, either in mixture or in superimposed layers; the same was reported by Guichard and Guineau on Roman age paintings (2002)."

     Line 535- The authors say madder is not used in Etruria, but then the sarcophagus of the Amazons is cited, which is from Etruscan Tarquinia. The chapter from that book (Il ‘sarcofago delle Amazzoni’ di Tarquinia), plus its pages needs to be cited rather than the overall book.

     There seems to be some confusion on geography. Amid line 532, the authors refer to what is happening artistically in South Italy and then suddenly Macedonia appears. Since the paper deals with Apulian tombs, the authors need to clearly state that the results in relation to other northern Italian works of the same period, and then perhaps comment on the relationship to the wider Mediterranean world. 

     Line 540; "the use of imported pigments such as cinnabar or difficult to work with such as madder lake.​​"  Cinnabar was also difficult to work with insofar as special handling was required for fear of blackening.

Trivial Errors and Suggested Improvements

     The authors alternate between use of BC and BCE.  I would use BCE exclusively.  Page 1-

     l. 17: "In four tomb sthe pictorial layer was applied"-

     l. 65- missing parenthesis

     l. 353: Pliny's text is listed alternately as incorrect NH, but also correctly listed as HN. Change all to HN.

     Hematite is alternatively spelled haematite.  Change all to hematite.

     Pg 14 unclear English: "As is evident from the SEM image in Figure 7, between the madder lake pigment layer – thick about 20 microns – and the plaster, a Ca-based layer, absent in the different colored areas, is evident.​​"

     ll. 459-60: "thin thicknesses​​" to “thinness”

     L.631- extraneous N in title

I would recommend the authors prepare a revised version of the article.

Author Response

All the suggestions/corrections made by the referee, whom we thank, have been taken on board and the manuscript amended

Reviewer 3 Report

For the article "Pigments and Techniques of Hellenistic Apulian Tomb Painting", I have just few minor concerns as follows:

The second paragraph of the Introduction part may be summarized in a Table showing the places of the tombs, dates and painting figures. It will be easier to follow all the information given in this paragraph. And a map showing all the locations mentioned may be added as a figure in the text.

In the part 2.2 (Techniques), what was the number of scans for ATR measurements? What was the deviation for EDS since the authors mention that they used mineral standards to check the accuracy of the analytical data?

In the caption of Fig. 9, are the acquisition parameters correct? They mention sample 2 but there is no sample 2 in the graph.

Overall, the study is well presented. It can be published after minor revisions.

Author Response

All suggestions/corrections made by the referee, whom we thank, have been taken on board and the manuscript amended (Figure 9 has been deleted, following the suggestion of referee 2).

Round 2

Reviewer 1 Report

The authors have properly addressed most of my comments and suggestions. The quality of the manuscript has improved.